# Development and performance evaluation of a solenoid valve assisted low-cost ventilator on gas exchange and respiratory mechanics in a porcine model

Daniel Shyu[1], Peter Bliss[2], Alexander Adams[3], Roy Joseph Cho[4]*

**1** Department of Medicine, University of Minnesota, Minneapolis, Minnesota, United States of America,
**2** Independent Researcher, Philips Respironics, Minneapolis, Minnesota, United States of America,
**3** Department of Respiratory Therapy, Hennepin County Medical Center, Minneapolis, Minnesota, United States of America, **4** Department of Medicine, Section of Interventional Pulmonology, University of Minnesota, Minneapolis, Minnesota, United States of America

* choxx548@umn.edu

## Abstract

### Introduction

During the COVID-19 pandemic, ventilator shortages necessitated the development of new, low-cost ventilator designs. The fundamental requirements of a ventilator include precise gas delivery, rapid adjustments, durability, and user-friendliness, often achieved through solenoid valves. However, few solenoid-valve assisted low-cost ventilator (LCV) designs have been published, and gas exchange evaluation during LCV testing is lacking. This study describes the development and performance evaluation of a solenoid-valve assisted low-cost ventilator (SV-LCV) in vitro and in vivo, focusing on gas exchange and respiratory mechanics.

### Methods

The SV-LCV, a fully open ventilator device, was developed with comprehensive hardware and design documentation, utilizing solenoid valves for gas delivery regulation. Lung simulator testing calibrated tidal volumes at specified inspiratory and expiratory times, followed by in vivo testing in a porcine model to compare SV-LCV performance with a conventional ventilator.

### Results

The SV-LCV closely matched the control ventilator's respiratory profile and gas exchange across all test cycles. Lung simulator testing revealed direct effects of compliance and resistance changes on peak pressures and tidal volumes, with no significant changes in respiratory rate. In vivo testing demonstrated comparable gas exchange parameters between SV-LCV and conventional ventilator across all cycles. Specifically, in cycle 1, the SV-LCV showed arterial blood gas (ABG) results of pH 7.54, PCO2 34.5 mmHg, and PO2 91.7

**Data Availability Statement:** All relevant data are within the manuscript and its Supporting Information files.

**Funding:** University of Minnesota Emergency Response Grant 10K USD. The funders had no role in study design, data collection and analysis, decision to publish, or preparation of the manuscript

**Competing interests:** The authors have declared that no competing interests exist.

mmHg, compared to the control ventilator's ABG of pH 7.53, PCO2 37.1 mmHg, and PO2 134 mmHg. Cycle 2 exhibited ABG results of pH 7.53, PCO2 33.6 mmHg, and PO2 84.3 mmHg for SV-LCV, and pH 7.5, PCO2 34.2 mmHg, and PO2 93.5 mmHg for the control ventilator. Similarly, cycle 3 showed ABG results of pH 7.53, PCO2 32.1 mmHg, and PO2 127 mmHg for SV-LCV, and pH 7.5, PCO2 35.5 mmHg, and PO2 91.3 mmHg for the control ventilator.

## Conclusion

The SV-LCV provides similar gas exchange and respiratory mechanic profiles compared to a conventional ventilator. With a streamlined design and performance akin to commercially available ventilators, the SV-LCV presents a viable, readily available, and reliable short-term solution for overcoming ventilator supply shortages during crises.

## Introduction

The COVID-19 pandemic and its surges have exposed critical shortages in medical supplies, with one of the most concerning being the national supply of mechanical ventilators. Early on in the pandemic, considerable attention was given to this shortage given its direct effect on worse outcomes and many states and organizations developed strategies for rationing ventilators with triaging based on which patients were most likely to benefit [1]. As a result, the U.S. Department of Health and Human Services (HHS) and the Federal Emergency Management Agency (FEMA) engaged with healthcare systems, academic institutions, National Academies of Science, Engineering and Medicine, and large manufacturers to develop crisis standards to meet ventilator needs when resource capacity has been exceeded. To meet these growing demands, the HHS Assistant Secretary for Health and U.S. Surgeon General issued an open letter stressing the need to optimize use and allocation of mechanical ventilators and provide federal procurement to increase the strategic national stockpile capacity [2]. According to the MIT and widely circulated Institute for Health Metrics and Evaluation (IHME) ventilator models [3–5], an estimated 25,000 additional ventilators would be required to care for COVID-19 patients, which were roughly consistent with a 2015 study reporting 96,596 ICU beds in the United States with an allocation of about one ventilator per ICU bed in the United States [6]. To meet anticipated demand, emergency triage scenarios were invoked requiring, possibly, makeshift resuscitator ventilator use [7] or ventilating 2–4 patients with one ventilator [8]. In preparation, most hospitals cancelled elective procedures.

To meet the anticipated emergent needs during the COVID-19 pandemic, the United States, using the Defense Production Act, ordered 200,000 ventilators from eleven different manufacturers, all with different capabilities and design. There was a relative increase of adult mechanical ventilators from 2019 to 2020 of 31.5%. Given both the lack of supply and, at the time, prohibitive cost of ventilators, the United States Food and Drug Administration (FDA) enacted an emergency use authorization (EUA) for ventilators and multiple groups took on the challenge of creating new designs [9].

A capable ventilator must deliver precise volumes, rates, positive end-expiratory pressure (PEEP), allow for rapid adjustments, outline reasonable alarm limits, display pressure/flow graphics, have weaning options, and perform for days or weeks with rare error or failure possibility. The Association for the Advancement of Medical Instrumentation (AAMI) and FDA outlined design guidance and standards for these emergency use ventilators [9,10].

Furthermore, ventilators must be designed with practical application in mind, as clinical staff (physicians, nursing staff, respiratory therapists) would have to be comfortable and capable in managing and troubleshooting the ventilator. With these requirements in mind, several designs were published, including the University of Minnesota Ventilator, solenoid-based LCVs, O2U, and the People's Ventilator Project [7,11–14]. These varied in how oxygen was delivered (pressure-controlled or volume-controlled ventilation), cost, flexibility in the clinical setting, and testing (in vitro vs. animal in vivo).

As the pandemic progressed, the proportion on COVID-19 patients requiring hospitalization, intensive care, and mechanical ventilation in the ICU has stabilized throughout the surges [15,16]. As a result, the initial concern of ventilator shortage has decreased; however, with recent occurrences of other natural disasters, threat of terrorism, and possible future pandemics, there continues to be a need for mass-casualty mechanical ventilation beyond the national stockpile program [16]. Our group at the University of Minnesota aimed to develop a portable, solenoid-valve assisted low-cost ventilator (SV-LCV) that would provide modern-ventilator features including delivery of precise volume, rates, PEEP, allow for rapid adjustments, outline reasonable alarm limits, display pressure/flow graphics, and operate consistently over a range of settings. These design requirements are one of the reasons why a solenoid valve-based control system has been the design of choice for modern ventilators. Aside from the emergency applications for a LCV, other avenues for LCV use in underserved medical areas including emergency, surgery, and transport of patients. To our knowledge, there have only been two solenoid valve LCV designs published; moreover, no LCV testing has demonstrated gas exchange performance. The purpose of this study is to develop and test the performance of our SV-LCV in both gas exchange and respiratory mechanic profile using a porcine model. We have also provided detailed instruction on the hardware and build schematics of our SV-LCV.

## Methods

The study was conducted at the University of Minnesota's Advanced Preclinical Imaging Center (APIC) and Interventional Pulmonary Airway Lab. The protocol was approved by the Institutional Animal Care and Use Committee of the University of Minnesota, Minneapolis MN.

### Electrical and mechanical schematics

The details of the LCV with schematics are shown in Figs 1 and 2. The LCV utilizes a compressed air or blended gas source. Triggering is mandatory time cycled, the delivery mode is a mandatory volume, constant flow with pressure plateau and positive end expiratory pressure (PEEP) can be applied. The LCV was designed to incorporate as many of the safety features required of a critical care ventilator as possible in a low-cost manner. The regulatory guidance documents IEC 60601–1 for Medical Electrical Equipment and its related standards, most notably ISO 80601-2-12 for Critical Care Ventilators provide many of the requirements. These include an anti-asphyxia valve, mechanical over pressure relief valve, and alarms for low pressure (implying circuit disconnect), high pressure and insufficient flow.

Inlet gas passes through a filter to a pressure regulator (Airtrol Components R-910) which allows operation across a range of inlet pressures. Flow to the patient is switched with a two-way solenoid valve (Parker Hannifin MX7). This valve is opened either for the duration of the set inspiratory time or closed when a set plateau pressure is achieved. A mechanical needle valve sets the delivery flow rate. A fixed restriction creates a pressure drop roughly proportional to the flow rate, which is measured with a differential pressure transducer (Honeywell SSC) and an identical transducer monitors the pressure at the patient port. The patient port also includes the anti-asphyxia valve and overpressure relief valve.

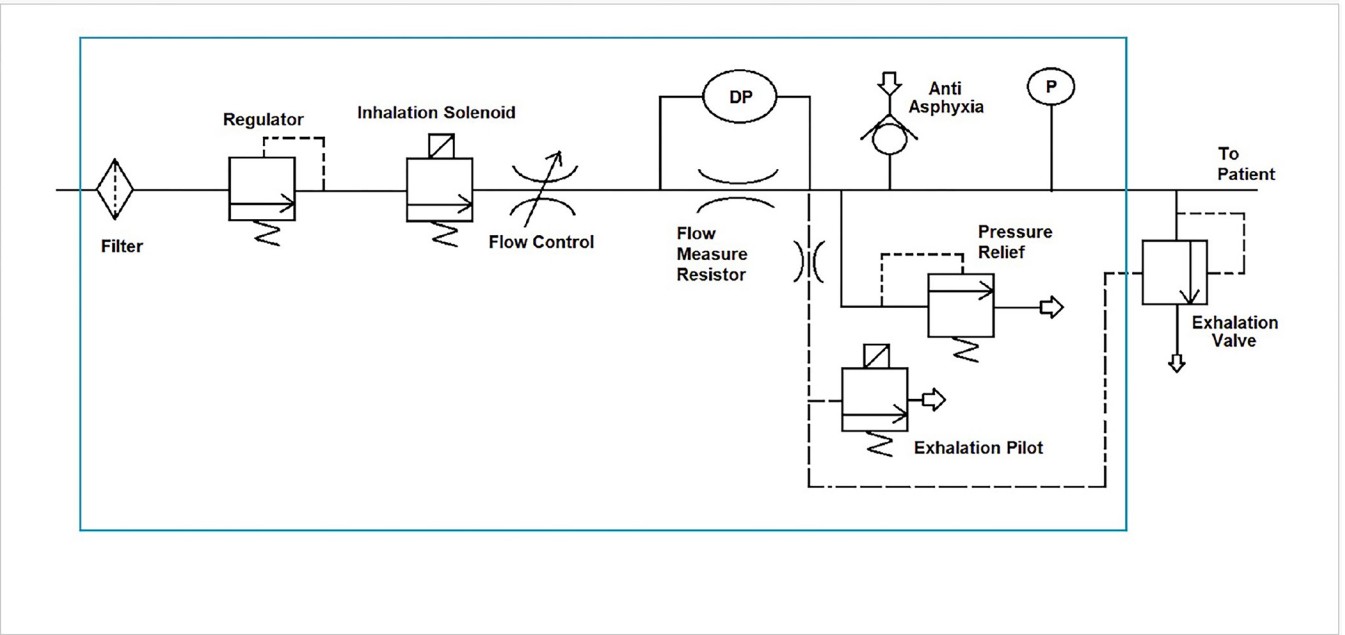

**Fig 1. LCV setup.** Left photo illustrates A) LCV ventilator, B) Software, and C) PEEP pop-off valve. Right photo represents the solenoid and A) flow control valve.

A single lumen ventilator circuit (Air Life) includes a pneumatically actuated exhalation valve proximal to the patient. The valve is pressurized to the closed position from positive pressure at the outlet of the patient gas delivery solenoid. A solenoid valve (Parker Hannifin MX7) will bleed gas away to open the exhalation valve and can be closed when a desired pressure value is reached to maintain PEEP.

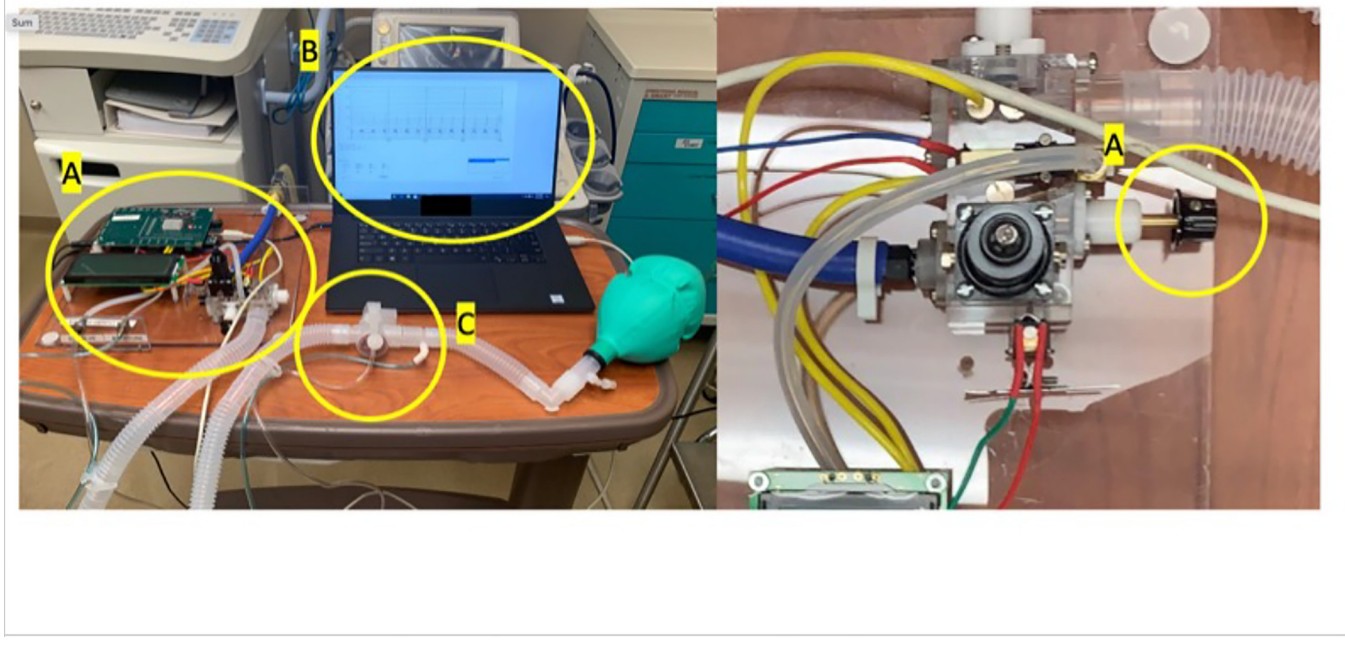

**Fig 2. LCV schematic.**

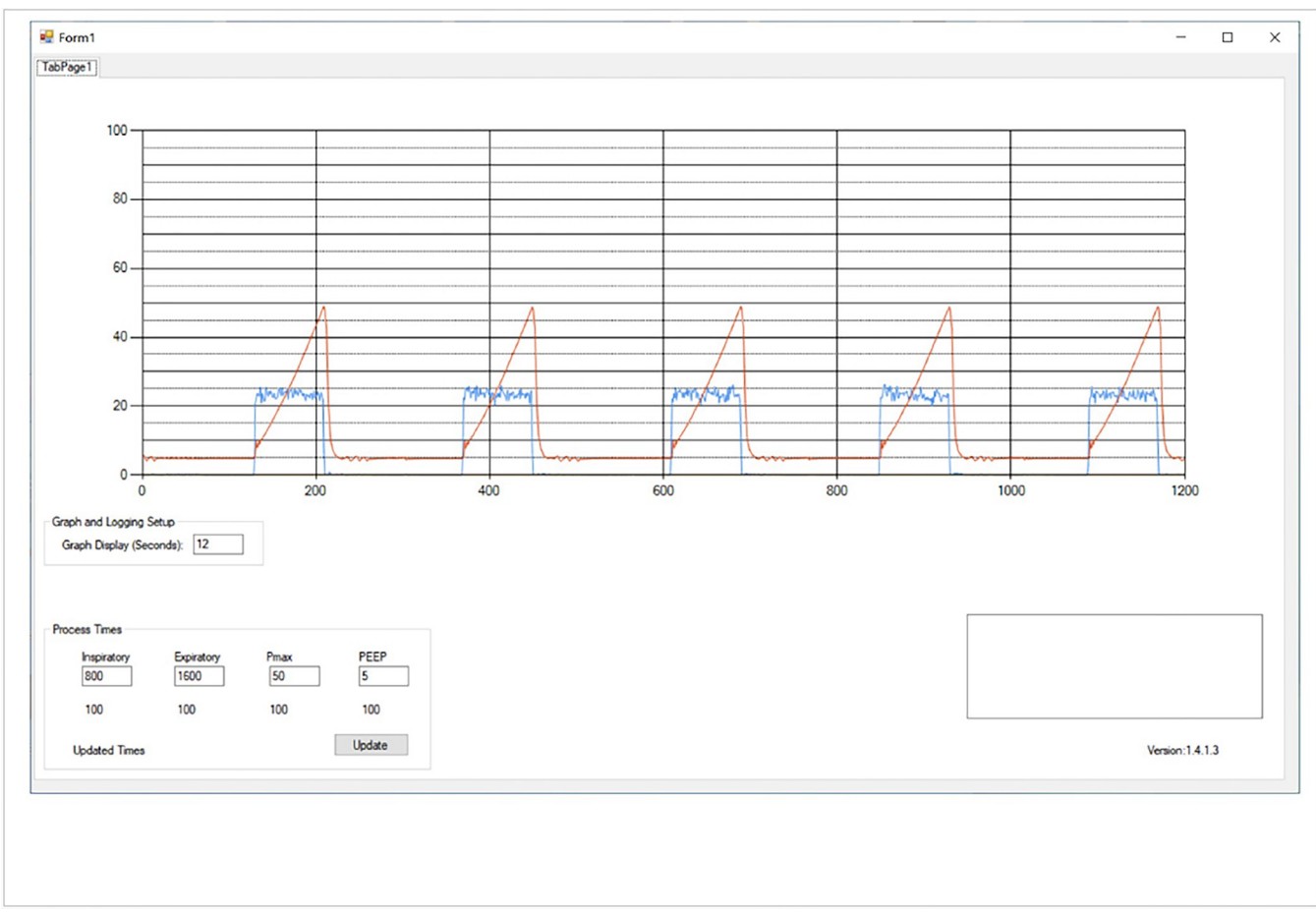

**Fig 3. Software graphics of the LCV.** Te, Ti, Pmax alarm and PEEP can be entered as shown. The waveform displays both pressure (red) and flow (blue) curves.

Control of the device is achieved with custom software written using Microsoft Visual Studio and a microcontroller circuit board (Arduino Mega). The software runs and displays data and waveforms on a personal computer (see Fig 3). In the software, inhalation and exhalation Times, plateau pressure and PEEP can be set.

## Lung simulator testing

A breathing simulator (Hans Rudolph Series 1101, Inc. Missouri USA) was used to test the LCV output including tidal volume (VT), respiratory rate (RR), PEEP, and peak pressures. The Hans breathing simulator (HBS) was set to resemble normal lung condition: resistance 5 cm/L/s, compliance 100 mL/cmH2O, breath rate 24 bpm, amplitude 0 cmH2O to maintain a passive state, % inhale of 33% to establish I:E (inspiratory to expiratory) ratio of 1:2, and target volume of 3000 mL. Table 1 illustrates how the software adjusted the inspiratory and expiratory times to set a respiratory rate for a given ratio. Tidal volume was calibrated using the HBS and varying flow rates from the device. The compliance, resistance, peak flow (Pkf, LPM), peak airway pressure (Pkp, cmH2O), PEEP (cmH2O), RR (breaths per minute), inspiratory time (Ti, sec), expiratory time (Te, sec), and VT (mL) were recorded for every condition tested. Each condition was repeated five times with results displayed as mean or mean ± standard deviation (SD).

**Table 1. Inspiratory (Ti) and Expiratory time (Te) tables to set a respiratory rate.**

| Inspiratory and Expiratory Times in mS | | | Ti = 60/Resp. Rate x Insp. Fraction x 1000 | | | | | | | |
|---|---|---|---|---|---|---|---|---|---|---|
| | | | Te = 60/Resp. Rate x Ex. Fraction x 1000 | | | | | | | |
| | | | | | I:E Ratio | | | | | |
| Resp. Rate | 1:2 | | 1:1 | | | 2:1 | | 1:3 | |
| | Ti | Te | Ti | Te | | Ti | Te | Ti | Te |
| 10 | 2000 | 4000 | 3000 | 3000 | | 4000 | 2000 | 1500 | 4500 |
| 11 | 1818 | 3637 | 2727 | 2727 | | 3637 | 1818 | 1364 | 4091 |
| 12 | 1667 | 3334 | 2500 | 2500 | | 3334 | 1667 | 1250 | 3750 |
| 13 | 1538 | 3077 | 2308 | 2308 | | 3077 | 1538 | 1154 | 3462 |
| 14 | 1428 | 2857 | 2143 | 2143 | | 2857 | 1428 | 1071 | 3214 |
| 15 | 1333 | 2667 | 2000 | 2000 | | 2667 | 1333 | 1000 | 3000 |
| 16 | 1250 | 2500 | 1875 | 1875 | | 2500 | 1250 | 938 | 2813 |
| 17 | 1176 | 2353 | 1765 | 1765 | | 2353 | 1176 | 882 | 2647 |
| 18 | 1111 | 2222 | 1667 | 1667 | | 2222 | 1111 | 833 | 2500 |
| 19 | 1053 | 2105 | 1579 | 1579 | | 2105 | 1053 | 789 | 2368 |
| 20 | 1000 | 2000 | 1500 | 1500 | | 2000 | 1000 | 750 | 2250 |
| 21 | 952 | 1905 | 1429 | 1429 | | 1905 | 952 | 714 | 2143 |
| 22 | 909 | 1818 | 1364 | 1364 | | 1818 | 909 | 682 | 2045 |
| 23 | 869 | 1739 | 1304 | 1304 | | 1739 | 869 | 652 | 1957 |
| 24 | 833 | 1667 | 1250 | 1250 | | 1667 | 833 | 625 | 1875 |
| 25 | 800 | 1600 | 1200 | 1200 | | 1600 | 800 | 600 | 1800 |
| 26 | 769 | 1539 | 1154 | 1154 | | 1539 | 769 | 577 | 1731 |
| 27 | 741 | 1482 | 1111 | 1111 | | 1482 | 741 | 556 | 1667 |
| 28 | 714 | 1429 | 1071 | 1071 | | 1429 | 714 | 536 | 1607 |
| 29 | 690 | 1379 | 1034 | 1034 | | 1379 | 690 | 517 | 1552 |
| 30 | 667 | 1333 | 1000 | 1000 | | 1333 | 667 | 500 | 1500 |

Ti = 60/Resp. Rate x Insp. Fraction x 1000.

Te = 60/Resp. Rate x Ex. Fraction x 1000.

## Anesthesia protocol

Swine will be fasted overnight and have unlimited access to water. For initial sedation, animal will receive ketamine (20–30 mg/kg, IM) and xylazine (1–3 mg/kg). Animals are transferred from their holding pen to the surgical suite. If further sedation is needed to safely transport animal to OR they will be masked down with Isoflurane via a portable anesthesia machine. Once in OR, an IV will be placed in the ear vein and they will be intubated with an appropriately sized endotracheal tube. If needed, propofol (1–2 mg/kg, IV) may be used as supplemental induction sedative. Anesthesia with isoflurane will be administered to effect (0.5–4%). Ventilation will be with a volume-controlled ventilator (Narkomed 2A, Drager) set to a tidal volume of 10 ml/kg and a respiratory rate to keep oxygen saturation above 95% and end tidal carbon dioxide (ETCO2) between 35–40 mmHg. Hemodynamics, electrocardiogram, ETCO2 and oxygen saturation will be continuously monitored with a patient monitor and recorded.

A modified Seldinger approach (under ultrasound guidance) will be used to cannulate the right or left femoral artery to measure invasive blood pressure and the right jugular vein will used to administer fluids and emergency medications. Body temperature will be maintained between 37-38C and measured with an esophageal temperature probe.

At beginning of prep, animal will receive Saline (1L, IV), Carprofen (2–3 mg/kg, SQ). Heparin (5000 IU, IV) will be administered when surgical access is complete.

## Ventilator protocol

A 60-kg farm animal pig was used to test the SV-LCV under real-world circumstances. Three cycles were performed with the SV-LCV and control ventilator (Drager Narkomed GS). Each cycle duration was 20-minutes of uninterrupted mechanical ventilation. The control ventilator settings were fraction of inspired oxygen ($FiO_2$) 20%, VT 520 cc, RR 20 BPM, PEEP 5 $cmH_2O$, and Flow rate of 60 LPM. After each 10-min cycle, an arterial blood gas (ABG) was drawn to compare systemic ventilation with the control ventilator and SV-LCV. Each ABG consisted of pH, partial pressure of $CO_2$ ($pCO_2$) and partial pressure of $O_2$ ($pO_2$) within the arterial circulation. The ABG was obtained from a central access catheter that was inserted into the femoral artery of the pig.

## Statistics

Descriptive statistics (means and standard deviations for continuous variables; counts and proportions for categorical variables) were used to summarize the study measurements. P-values of $< 0.05$ were considered statistically significant. All analyses were carried out using SAS V9.3 (SAS Institute Inc., Cary, NC).

## Results

### Lung simulator test

The SV-LCV's performance was tested using the HBS under two conditions (see Table 2): 1) varying compliances with fixed resistance and 2) varying resistances with fixed compliance. At a given resistance, the compliance changes had a direct effect on Pkp (see Table 2). There was no significant change in RR however VT did decrease with worsening compliance as expected (highest 547±8 mL for compliance of 100 $mL/cmH_2O$, lowest 430±2 mL for compliance of 10 $ml/cmH_2O$). At a given compliance, the resistance changes had a direct effect on Pkp (lowest 15 $cmH_2O$ with resistance of 5 cm/L/s, highest 34 $cmH_2O$ resistance of 50 cml/L/s). There was no

**Table 2. LCV Response in RR and VT with varying compliance and resistance using a lung simulator.** Flow rate, Ti, Te, Compliance and Resistance are the control variables. In this series, Compliance and Resistance varied to determine effect on Pkp, RR and VT.

| Flow valve (L/min) | Ti (ms) | Te (ms) | Compliance (ml/cmH2O) | Resistance (cm/l/s) | PKp (cmH2O) | RR | VT (mL) |
|---|---|---|---|---|---|---|---|
| 30 | 2000 | 4000 | 100 | 5 | 15 | 10±1 | 547±8 |
| 30 | 2000 | 4000 | 75 | 5 | 15 | 10±2 | 543±5 |
| 30 | 2000 | 4000 | 50 | 5 | 17 | 11±1 | 537±7 |
| 30 | 2000 | 4000 | 25 | 5 | 25 | 10±1 | 505±5 |
| 30 | 2000 | 4000 | 10 | 5 | 54 | 10±1 | 430±2 |
| 30 | 2000 | 4000 | 100 | 5 | 15 | 10±1 | 547±5 |
| 30 | 2000 | 4000 | 100 | 7.5 | 17 | 10±1 | 530.5±5 |
| 30 | 2000 | 4000 | 100 | 10 | 20 | 11±1 | 523.5±5 |
| 30 | 2000 | 4000 | 100 | 20 | 29 | 12±1 | 491±10 |
| 30 | 2000 | 4000 | 100 | 50 | 34 | 10±1 | 433±10 |
| | | | | | | | |

Ti = Inspiratory time, Te = Expiratory time, PKp = Peak airway pressure, RR = Respiratory rate, VT = Tidal volume.

significant change in RR however VT did decrease slightly with worsening resistance as expected (highest 547±5 mL for resistance of 5 cm/L/s, lowest 433±10 mL for resistance of 50 cm/L/s).

## Animal test

The SV-LCV was calibrated to achieve a VT 500 cc and a RR of 20 BPM with Ti 750 ms, Te 2200 ms. real-time VT was confirmed with an in-line volume meter. The SV-LCV matched the control ventilator's VT, RR, PEEP and FiO2%. Table 3 shows the ventilator parameters and ABG results for each of the three cycles, comparing the control ventilator with the SV-LCV. In all three cycles, the SV-LCV closely ventilated and oxygenated the porcine model in comparison to the control ventilator (see Table 3). The VT and RR on the SV-LCV were programmed independently to match the minute ventilation of the control ventilator. The pH, PCO2 and PO2 was well maintained in all three cycles using the SV-LCV.

## Discussion

Early in the COVID-19 pandemic, there was significant concern for shortage of mechanical ventilators and that triaging patients would be needed. To attempt to address the supply concern, new LCV designs were developed. Here, we describe the development and performance of our SV-LCV design in both an in vitro and in vivo testing scenario that could be used as a framework for solenoid-based LCVs in the future if the demand outpaces the supply again.

Ventilator designs that received FDA Emergency Use Authorizations varied in design, ranging from 3D printing a device to repurpose bag valve mask manual resuscitators to using a piston-cylinder model, or using pre-pressurized gas with valves to control volume and pressure of air delivered to the patient [11–15]. Piston-cylinder designs such as the MADVent, use

**Table 3. Three cycles performed using a conventional ventilator vs LCV.** The control ventilator was set at VT 520cc, RR 20 bpm, PEEP 5 cmH2O, and Flow 60 LPM. The LCV was set at PEEP = 5 cmH2O, Ti 750 ms, and Te 2200 ms.

|  |  | Cycle 1 |  | Cycle 2 |  | Cycle 3 |  |
| --- | --- | --- | --- | --- | --- | --- | --- |
| **Control Ventilator** | FiO2 (%) | 21 | FiO2 (%) | 21 | FiO2 (%) | 21 |  |
|  | VT | 520 | VT | 520 | VT | 520 |  |
|  | RR | 20 | RR | 20 | RR | 20 |  |
|  | PEEP | 5 | PEEP | 5 | PEEP | 5 |  |
|  | Flow | 60 | Flow | 60 | Flow | 60 |  |
|  | PH | 7.52 | PH | 7.5 | PH | 7.5 |  |
|  | pCO2 | 37.1 | pCO2 | 34.2 | pCO2 | 35.5 |  |
|  | PO2 | 134 | PO2 | 93.5 | PO2 | 91.3 |  |
| **LCV** | FiO2 (%) | 21 | FiO2 (%) | 21 | FiO2 (%) | 21 |  |
|  | Ti | 750 | Ti | 750 | Ti | 750 |  |
|  | Te | 2200 | Te | 2200 | Te | 2200 |  |
|  | VT | 500* | VT | 500* | VT | 500* |  |
|  | PEEP | 5 | PEEP | 5 | PEEP | 5 |  |
|  | RR | 20* | RR | 20* | RR | 20* |  |
|  | PH | 7.538 | PH | 7.53 | PH | 7.53 |  |
|  | pCO2 | 34.5 | pCO2 | 33.6 | pCO2 | 32.1 |  |
|  | PO2 | 91.7 | PO2 | 84.3 | PO2 | 127 |  |

*The flow valve was tuned to achieve VT 500 cc using an inline volume meter. The Ti and Te values acheived a RR 20.

Ti = Inspiratory time (ms), Te = Expiratory time (ms), PKp = Peak airway pressure (cmH2O), RR = Respiratory rate (BPM), VT = Tidal volume (cc), PEEP = Postive end-expiratory pressure, pCO2 = partial pressure CO2, PO2 = partial pressure O2, FiO2 = Inspiratory O2%.

a motor to compress a resuscitation bag optimized to a specific tidal volume [12]. Although a simple design, the main issue is compliance of the resuscitation bag overtime resulting in a potential variation in delivered minute ventilation. Recently, the O2U ventilator used pressurized medical gases with time limited flow interruption to determine the rate and volume, a design that can only be used where medical grade pressurized gas is available. In addition, the PVP1 utilized a similar design to this LCV but did not run any in vivo testing. In the literature, there are only two published reports of solenoid valve-based LCVs [13,14]. The main advantages for a solenoid valve ventilator compared to bag valve, piston and mechanical devices include fail-safe design (open, close position), high life cycle, fast-acting, and can operate under AC or DC power. Here, our SV-LCV combines several of the elements in a simple design with fidelity noted in both in vitro and in vivo testing. With the use of a pressure regulator, solenoid valve, this design can use either compressed air or a blended gas source. Uniquely, we have provided gas exchange data in comparison with a conventional ventilator during performance testing in a porcine model. Although respiratory mechanics is an important measure of success with these LCVs, physiological outcomes including ventilation and oxygenation data affirms the ultimate goal of these devices, which is to provide reliable, consistent ventilation and oxygenation to patients in times of crisis [15–18]. To our knowledge, our study is the first to provide this information.

Testing was done both with a human lung simulator and with a porcine model in comparison with a commercially available ventilator. Under normal lung conditions, the SV-LCV performed reasonably well maintaining a near constant minute ventilation at a 1:2 I:E ratio with no significant changes in RR or VT. However, when lung compliance and resistance varied, we observed a reduction in minute ventilation (mainly VT) and increase in airway pressures accordingly. When this occurred, VT was able to be maintained over the flow rate ranges using the control valve. Therefore, under conditions where compliance and resistance vary, the actual delivered VT per cycle would require frequent monitoring for adequate ventilation and airway pressures using expiratory volume manometers to safely ventilate. The addition of the pop-off valve helped ensure a limit to the peak pressure. Although this level of monitoring would be difficult to maintain throughout the entire course of a patient's acute illness, it would be feasible to use the SV-LCV for the short-term (e.g., less than 12-hours) during max capacity and when plans for reallocation of a conventional ventilator are underway. In the animal model, the SV-LCV was compared to the control ventilator in a porcine model with ABGs collected to assess for quality of ventilation. Each of the three 10-minute cycles showed that the SV-LCV was able to obtain comparable pH, pCO2, and pO2. These limited trials show that the SV-LCV can provide adequate ventilation for gas exchange compared to standard modern ventilators used in the OR. This degree of physiological testing is novel to this study.

This study has several limitations. For the simulated testing, we tested changes in only compliance and resistance under passive conditions; therefore, the effect of spontaneous effort was not evaluated. Additionally, we did not alter the I:E ratio (and consequentially respiratory rate) so performance along low and high-rate ventilation is unknown over a 10-min cycle. Similarly, the performance under high peak pressures has not been evaluated. In the animal model, only three trials were done, each lasting for 10-minutes, much shorter than the time frame where a ventilator would need to perform. In addition, this test only evaluated for gas exchange and did not address the possible side effects of ventilation, such as barotrauma, atelectasis, etc., that can occur. Aside from these limitations, we believe that this study demonstrates the indications and limitations of using an SV-LCV and can serve as a basis for future work.

## Conclusion

We conducted extensive testing on our SV-LCV across various conditions to assess its performance and, crucially, its suitability for human use. We have made available detailed instructions and schematics for public scrutiny and evaluation. Our testing involved comparing the respiratory mechanical profile and gas exchange of our SV-LCV with that of a conventional ventilator, using a live porcine model. The results indicate that our ventilator achieves similar outcomes to conventional models. However, it is important to note that our device cannot fully replace full-featured ventilators. Instead, it presents itself as a viable option when ventilator capacity is stretched, and short-term support is necessary to buy time while reallocation plans for ventilators are executed. Given its consistent performance under typical lung conditions, our ventilator may find its niche in patients requiring lesser ventilatory support. This could allow full-featured ventilators to be prioritized for those experiencing acute declines in ventilation and oxygenation.

## Author Contributions

**Conceptualization:** Peter Bliss, Alexander Adams, Roy Joseph Cho.

**Data curation:** Roy Joseph Cho.

**Formal analysis:** Daniel Shyu, Peter Bliss, Alexander Adams, Roy Joseph Cho.

**Funding acquisition:** Roy Joseph Cho.

**Investigation:** Peter Bliss, Roy Joseph Cho.

**Methodology:** Peter Bliss, Roy Joseph Cho.

**Project administration:** Peter Bliss, Roy Joseph Cho.

**Resources:** Peter Bliss, Roy Joseph Cho.

**Software:** Roy Joseph Cho.

**Supervision:** Roy Joseph Cho.

**Validation:** Peter Bliss, Roy Joseph Cho.

**Visualization:** Roy Joseph Cho.

**Writing – original draft:** Daniel Shyu, Roy Joseph Cho.

**Writing – review & editing:** Daniel Shyu, Peter Bliss, Alexander Adams, Roy Joseph Cho.

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
