## [Decision Letter · Decision Letter 0]

19 Jul 2023

PONE-D-23-12462To Test the Performance of an Open-Source, Low-Cost Ventilator in a Porcine Model.PLOS ONE

Dear Dr. Cho,

Thank you for submitting your manuscript to PLOS ONE. After careful consideration, we feel that it has merit but does not fully meet PLOS ONE’s publication criteria as it currently stands. Therefore, we invite you to submit a revised version of the manuscript that addresses the points raised during the review process.

We look forward to receiving your revised manuscript.

Kind regards,

Mattias Günther

Academic Editor

PLOS ONE

2. As part of your revision, please complete and submit a copy of the Full ARRIVE 2.0 Guidelines checklist, a document that aims to improve experimental reporting and reproducibility of animal studies for purposes of post-publication data analysis and reproducibility: https://arriveguidelines.org/sites/arrive/files/documents/Author%20Checklist%20-%20Full.pdf Please include your completed checklist as a Supporting Information file. Note that if your paper is accepted for publication, this checklist will be published as part of your article.

3. To comply with PLOS ONE submissions requirements, in your Methods section, please provide additional information regarding the experiments involving animals and ensure you have included details on (1) methods of sacrifice, (2) methods of anesthesia and/or analgesia, and (3) efforts to alleviate suffering.

7. Please include your tables as part of your main manuscript and remove the individual files. Please note that supplementary tables (should remain/ be uploaded) as separate "supporting information" files

Reviewers' comments:

Reviewer's Responses to Questions

**Comments to the Author**

1. Is the manuscript technically sound, and do the data support the conclusions?

Reviewer #1: No

Reviewer #2: Yes

2. Has the statistical analysis been performed appropriately and rigorously? 

Reviewer #1: No

Reviewer #2: I Don't Know

3. Have the authors made all data underlying the findings in their manuscript fully available?

Reviewer #1: Yes

Reviewer #2: Yes

4. Is the manuscript presented in an intelligible fashion and written in standard English?

Reviewer #1: Yes

Reviewer #2: Yes

5. Review Comments to the Author

Reviewer #1: I am a little unclear what the author's expectations were for this article. They claim an open source design, one that is not thoroughly given or expanded upon. Additionally, the timelines of this topic is misaligned with the reality of current health states. The federal government has removed the state of emergency over COVID-19 and there have been several papers on this or a similar topic.

Additionally, proposing an "open-source" medical instrument like a ventilator is a very precarious position to take as the amount of safety concerns was also something brought up when teams were in the midst of the pandemic working on rapid response ventilators. There are several other papers exactly like this paper, including the ones cited and published in this journal that did porcine modeling as well. The novelty of this paper is therefore in question and the value that it adds to the journal and to the field I think is questionable. The reference list is woefully short for this kind of important paper and the depth of analysis, risk analysis, part sourcing is lacking. The animal study that was claimed was not detailed with any thoroughness or explanation. This paper is far too short and under developed for the important concepts introduced. There are many other papers, including ones the authors cite that cover this work in better detail, in PLOS ONE no less. Therefore I suggest the editor reject this submission.

Reviewer #2: The overall conclusion of this paper is well stated - this limited study describes the use of a low-cost ventilator that could be used in a crises scenario for short periods (<12hrs) of time. This paper serves as a basis for further study into the low-cost ventilator area of research.

Recommendations for Revision for Author:

There is no description of the anesthetic protocol or management for the porcine model. It is beneficial for the reader to understand what type of anesthetics were used for this study and what effect the anesthetics could have on the parameters evaluated.

Please make sure all acronyms are spelled out during their initial use.

For the lung simulator testing, the duration of testing for each condition is unclear.

Table 3 - Ti and Te for control ventilator were not provided in the chart. Are those values available? The legend of table 3 also appears cut off (no description of what PO2 is)

Recommended citation: King, W. P., Amos, J., Azer, M., Baker, D., Bashir, R., Best, C., ... & Wooldridge, A. R. (2020). Emergency ventilator for COVID-19. PLoS One, 15(12), e0244963.

6. PLOS authors have the option to publish the peer review history of their article (what does this mean?). If published, this will include your full peer review and any attached files.

Reviewer #1: No

Reviewer #2: No

---

## [Author Response · Author response to Decision Letter 0]

16 Feb 2024

Response to Referee and Editor Comments on the Paper:

“To Test the Performance of an Open-Source, Low-Cost Ventilator in a Porcine Model”

PLOS One Journal

PONE-D-23-12462

Response to Editor’s Comments:

For this revision, the Editor recommends that we consider the suggestions by the reviewer. This response note forms the main part of the requested accompanying letter. We also note here that we have been able to accommodate all the suggestions proposed by both reviewers. Below is the summary of the suggestions and our responses to them.

Reviewer #1

1. I am a little unclear what the author's expectations were for this article. They claim an open-source design, one that is not thoroughly given or expanded upon. Additionally, the timeline of this topic is misaligned with the reality of current health states. The federal government has removed the state of emergency over COVID-19 and there have been several papers on this or a similar topic. Additionally, proposing an "open source" medical instrument like a ventilator is a very precarious position to take as the amount of safety concerns was also something brought up when teams were in the midst of the pandemic working on rapid response ventilators. There are several other papers exactly like this paper, including the ones cited and published in this journal that did porcine modeling as well. The novelty of this paper is therefore in question and the value that it adds to the journal and to the field I think is questionable. The reference list is woefully short for this kind of important paper and the depth of analysis, risk analysis, part sourcing is lacking. The animal study that was claimed was not detailed with any thoroughness or explanation. This paper is far too short and underdeveloped for the important concepts introduced. There are many other papers, including ones the authors cite that cover this work in better detail, in PLOS ONE no less. Therefore, I suggest the editor reject this submission.

Response: Appreciate the comments and agree with the reviewed on the numerous reports for low-cost ventilators that have been produced during the COVID pandemic. However, we would like to provide reasoning and support on why our LCV study is unique compared to others that have been previously published. In a PubMed search for “low-cost ventilator”, there were 14 relevant published papers in the first 150 searches. Only two papers have a similar design using a solenoid valve to control flow rate. Furthermore, our study is the first in providing physiological gas exchange (e.g. arterial blood gas or ABG) data from an LCV ; in addition, compares the gas exchange data to that of a control ventilator in our animal model. Out of all the 14 LCV studies, physiological gas exchange has not been assessed to our knowledge. Of the two other solenoid valve designs, our LCV provides are more compact, portable design in comparison. Given the scarcity of data using solenoid valves in LCV and our gas exchange data, we feel that our study is novel and will add to the current literature. We also appreciate that more discussion could be added, and we have included that in our revision as well as including more references to support the discussions and our findings as well. The title, introduction and discussion paragraphs have been revised to highlight what stands out in our design in addition to emphasizing that we have physiological gas exchange data that has not been reported previously for LCV during the COVID pandemic. 

Reviewer #2 

1. There is no description of the anesthetic protocol or management for the porcine model. It is beneficial for the reader to understand what type of anesthetics were used for this study and what effect the anesthetics could have on the parameters evaluated.

Response: Agree. Added in the methods section. 

2. Please make sure all acronyms are spelled out during their initial use.

Response: Agree. Our manuscript has been revised. 

3. For the lung simulator testing, the duration of testing for each condition is unclear.

Response: Agree. Our manuscript has been revised. 

4. Table 3 - Ti and Te for control ventilator were not provided in the chart. Are those values available? The legend of table 3 also appears cut off (no description of what PO2 is)

Response: The Ti and Te chosen for this study resulted from calibration using our lung simulator. This provided us with the tidal volumes demonstrated in the porcine model. Legend for Table 3 has been corrected. 

5. Recommended citation: King, W. P., Amos, J., Azer, M., Baker, D., Bashir, R., Best, C., ... & Wooldridge, A. R. (2020). Emergency ventilator for COVID-19. PLoS One, 15(12), e0244963.

Response: Agree. Citation has been added.

---

## [Decision Letter · Decision Letter 1]

1 Apr 2024

PONE-D-23-12462R1Development and Performance Evaluation of a Solenoid Valve Assisted Low-Cost Ventilator on Gas Exchange and Respiratory Mechanics in a Porcine Model.PLOS ONE

Dear Dr. Cho,

Thank you for submitting your manuscript to PLOS ONE. After careful consideration, we feel that it has merit but does not fully meet PLOS ONE’s publication criteria as it currently stands. Therefore, we invite you to submit a revised version of the manuscript that addresses the points raised during the review process.

We look forward to receiving your revised manuscript.

Kind regards,

Mattias Günther

Academic Editor

PLOS ONE

Journal Requirements:

Reviewers' comments:

Reviewer's Responses to Questions

**Comments to the Author**

1. If the authors have adequately addressed your comments raised in a previous round of review and you feel that this manuscript is now acceptable for publication, you may indicate that here to bypass the “Comments to the Author” section, enter your conflict of interest statement in the “Confidential to Editor” section, and submit your "Accept" recommendation.

Reviewer #3: (No Response)

2. Is the manuscript technically sound, and do the data support the conclusions?

Reviewer #3: Yes

3. Has the statistical analysis been performed appropriately and rigorously? 

Reviewer #3: Yes

4. Have the authors made all data underlying the findings in their manuscript fully available?

Reviewer #3: Yes

5. Is the manuscript presented in an intelligible fashion and written in standard English?

Reviewer #3: Yes

6. Review Comments to the Author

Reviewer #3: The paper, “Development and Performance Evaluation of a Solenoid Valve Assisted Low-Cost Ventilator on Gas Exchange and Respiratory Mechanics in a Porcine Model.” has been presented. Technically, the article presents good scientific information and may be accepted for publication after clarifying some unclear and/or missing points:

1. Write the applications in the introduction section.

2. Add quantitative results in the abstract.

3. Provide references to support your results.

4. The results section needed to be improved as it seems to be a laboratory report rather than a scientific investigation.

5. Reconstructed the conclusion part.

7. PLOS authors have the option to publish the peer review history of their article (what does this mean?). If published, this will include your full peer review and any attached files.

Reviewer #3: **Yes: **Dr. Rajkumar Sivanraju

---

## [Author Response · Author response to Decision Letter 1]

18 Apr 2024

Figure 2 was revised and the "Dell" Logo is now covered and blacked out. Thank you

---

## [Decision Letter · Decision Letter 2]

25 Apr 2024

Development and Performance Evaluation of a Solenoid Valve Assisted Low-Cost Ventilator on Gas Exchange and Respiratory Mechanics in a Porcine Model.

PONE-D-23-12462R2

Dear Dr. Cho,

We’re pleased to inform you that your manuscript has been judged scientifically suitable for publication and will be formally accepted for publication once it meets all outstanding technical requirements.

Kind regards,

Mattias Günther

Academic Editor

PLOS ONE

Additional Editor Comments (optional):

Reviewers' comments:

Reviewer's Responses to Questions

**Comments to the Author**

1. If the authors have adequately addressed your comments raised in a previous round of review and you feel that this manuscript is now acceptable for publication, you may indicate that here to bypass the “Comments to the Author” section, enter your conflict of interest statement in the “Confidential to Editor” section, and submit your "Accept" recommendation.

Reviewer #3: All comments have been addressed

2. Is the manuscript technically sound, and do the data support the conclusions?

Reviewer #3: Yes

3. Has the statistical analysis been performed appropriately and rigorously? 

Reviewer #3: Yes

4. Have the authors made all data underlying the findings in their manuscript fully available?

Reviewer #3: Yes

5. Is the manuscript presented in an intelligible fashion and written in standard English?

Reviewer #3: Yes

6. Review Comments to the Author

Reviewer #3: The paper, “Development and Performance Evaluation of a Solenoid Valve Assisted Low-Cost Ventilator on Gas Exchange and Respiratory Mechanics in a Porcine Model” has been presented. The revised manuscript is improved a lot and the authors are responded well for the queries raised by me in my previous review. Now I accept this manuscript for publication.

7. PLOS authors have the option to publish the peer review history of their article (what does this mean?). If published, this will include your full peer review and any attached files.

Reviewer #3: **Yes: **Dr. Rajkumar Sivanraju

---

## [Editor Report · Acceptance letter]

7 May 2024

PONE-D-23-12462R2 

PLOS ONE

Dear Dr. Cho, 

I'm pleased to inform you that your manuscript has been deemed suitable for publication in PLOS ONE. Congratulations! Your manuscript is now being handed over to our production team.

Kind regards, 

on behalf of

Dr. Mattias Günther 

Academic Editor

PLOS ONE